

# Radar Reflectivity Factors Simulations of Ice Crystal Populations from In-Situ Observations for the Retrieval of Condensed Water Content in Tropical Mesoscale Convective Systems

Emmanuel. Fontaine[1,6*], Delphine. Leroy[1], Alfons. Schwarzenboeck[1], Julien. Delanoë[2], Alain. Protat[3],
Fabien. Dezitter[4], Alice. Grandin[4], John. Walter. Strapp[5], Lyle. Edward. Lilie[5].

[1]Université Blaise Pascal, Laboratoire de Météorologie Physique, Aubière, France

[2]Laboratoire Atmosphère, Milieux et Observations Spatiales, UVSQ, Guyancourt, France

[3]Center for Australian Weather and Climate Research, Melbourne, Australia

[4]Airbus, Toulouse, France

[5]Met Analytics, Toronto, Canada

[6]University of Reading, United Kingdom

*Correspondence to*: E. Fontaine (e.r.j.fontaine@reading.ac.uk)

**Abstract.** This study presents the evaluation of a technique to estimate cloud condensed water content (CWC) in tropical convection from airborne cloud radar reflectivity factors at 94GHz and in-situ measurements of particle size distributions (PSDs) and aspect ratios of ice crystal populations. The approach is to calculate the variability of 5 second average PSD CWCs and all possible solutions of corresponding m(D) relationships fulfilling the condition that the simulated radar reflectivity factors (T-matrix method) matches the measured reflectivity. For the reflectivity simulations, ice crystals were approximated

as oblate spheroids, without using a priori assumptions on the mass-size relationship of ice crystals. The CWC calculations demonstrate that individual CWC values are in the range ±32% of the average CWC value over all CWC solutions for the chosen 5s time intervals. In addition, during the airborne field campaign performed out of Darwin in 2014, as part of the international High Altitude Ice Crystals (HAIC) – High Ice Water Content (HIWC) projects, CWCs were measured directly with the new IKP-2 (isokinetic evaporator probe) instrument along with simultaneous particle imagery and radar reflectivity.

Averaged CWC retrieved from the radar reflectivity simulations are roughly 16% higher than the IKP-2 CWC measurements. The differences between the IKP-2 and PSD derived CWCs from the entire set of realistic m(D) solutions for T-matrix retrievals is found to be a function of the total number concentration of ice crystals. Consequently, a correction term is applied (as a function of total number concentration) that significantly improves the retrieved CWC. After correction, the retrieved CWC have a median error relative to measured values of only -1%.



# 1 Introduction

Clouds play a dominant role within the hydrological cycle on earth, and in its radiative transfer and heat balance. Thus, increasing knowledge and understanding of ice hydrometeor properties and processes is important for improving forecast and global climate models.

In order to derive cloud properties such as hydrometeor size distributions (ice or water), liquid and/or ice water content, ice particle shape (spherical, hexagonal, …), and precipitation rate, radar remote sensing (at different frequencies of 5.5GHz, 9.4GHz, 35GHz, 94GHz, …) is one of the most common measurement techniques. Radar reflectivity factors are an integral value of all backscattering cross sections from all hydrometeors within the radar sampling volume, which makes the radar a complex measurement device for estimating cloud properties.

The methodology applied in this study to simulate radar reflectivity factor is based on assumptions about individual cloud particle properties. If hydrometeors are droplets, then Mie solutions can be applied to the Maxwell equations; however, this is not the case for non-spherical ice crystals. Discrete dipole approximations (DDA; Draine and Flatau, 1994; Liu, 2008) can be used to calculate backscatter cross sections for complex shapes, and thus the more difficult problem of ice crystals. However, in order to apply DDA simulations to ice crystal radar reflectivity factors, a classification of ice hydrometeor habits is essential.

Unfortunately, more than 50% of ice crystal images from ground-based and airborne measurements are typically identified as irregular (do not fall within a unique habit classification) using automated and manual shape recognition techniques, , even when using very high resolution imaging such as from the Cloud Particle Imager (CPI, 2.3 µm resolution), (e.g. Mioche, 2010).

Hogan et al. (2011) used the oblate spheroid approximation to simulate radar reflectivity factors at 3GHz and 94GHz. For their

calculation of the ice fraction in horizontally oriented oblate spheroids these authors used a constant axis ratio of 0.6, and used the mass-size relationship from Brown and Francis (1995), originally from Locatelli and Hobbs(1974), to derive PSD mass distributions from number-concentration distributions. Even though a mass-size relationship for ice crystals with constant coefficients has been utilized, the Hogan et al. claimed minimal errors between measured and simulated radar reflectivity factors, even smaller than the calibration uncertainty of the cloud radar.

Fontaine et al. (2014) also used the oblate spheroid approximation for ice crystals in order to calculate condensed water content (CWC) in mesoscale convective systems (MCS), and Drigeard et al. (2015) used these findings to simulate radar reflectivity factors at 5.5GHz. Although simulations of radar reflectivity factors were in agreement with reflectivity observations at 94GHz and 5.5GHz, no direct bulk measurements of CWC were available to evaluate the retrieved CWCs.

During the High Altitude Ice Crystals (HAIC; Dezitter et al., 2013) / High Ice Water Content (HIWC; Strapp et al., 2016)

airborne field campaign out of Darwin, the Falcon-20 (F-20) from SAFIRE (Service des Avions Français Instrumentés pour la Recherche en Environnement) measured simultaneous cloud radar reflectivity factors (Z), CWC, PSD and aspect ratios of hydrometeors. This combination of cloud measurements allows for the evaluation of this method of simulating cloud radar





reflectivity using the oblate spheroid approximation, and the estimation of its accuracy in retrieving CWC in the ice phase part of mesoscale convective systems.

The next section of this paper presents the dataset of the first HAIC-HIWC airborne campaign and associated data processing. Subsequently, the cloud radar reflectivity simulation method (Fontaine et al. (2014)) also used in this study is briefly reviewed.

5 The CWC values, retrieved from the simulations are then compared to the collocated bulk measurements of CWC. Moreover, the study suggests correction functions for the CWC calculations (based on T-Matrix simulations of the reflectivity factors) as a function of temperature, largest size of hydrometeors in PSD, and ice crystal concentrations in MCS. A further section is dedicated to the estimation of possible uncertainties in the measurements leading to the presented CWC retrievals. Finally, the study ends with a discussion and conclusion section.

## 2 Data processing

In situ microphysical measurements and radar reflectivity data used in this study were provided by three types of instruments, that were mounted on-board the F-20 for the HAIC-HIWC campaign flown out of Darwin:

- The 94 GHz research cloud radar RASTA (Protat et al., 2009; Delanoe et al. 2013) measuring both cloud radar
reflectivity factors and 3D cloud dynamics below and above the flight level. The uncertainty on the measured reflectivity is about ± 1 dBZ (e.g., Protat et al. 2009).

- Two optical array probes (OAP), the 2D-Stereo probe from SPEC (Stratton Park Engineering Company Inc.) and the Precipitation Imaging Probe (PIP) from Droplet Measurement Technologies (DMT).

- The Isokinetic Evaporator Probe (IKP-2) providing direct CWC measurements. The IKP-2 is a second-generation
version of the prototype IKP (Davison et al. 2008), that was downsized for the F-20. The IKP-2 was developed to provide reliable measurements of CWC in deep convective clouds at temperatures colder than -10 C, up to at least $10g/m^3$ at 200 ms$^{-1}$, and with a target accuracy of 20%. The IKP-2 samples the cloud particles isokinetically, evaporates them, and measures the resulting humidity of the evaporated particles and background air. CWC (hereafter $CWC_{IKP}$ given in g/m$^3$) is then obtained by subtracting the water vapour background measurement from the IKP-2
total hygrometer signal. System accuracy estimates are better than 20% for CWC greater than 0.25 g/m$^3$ for temperatures lower than -10 C (Davison et al. 2016). Accuracy increases with decreasing temperature due to the exponential decrease in background humidity, which drives much of the IKP-2 error. At -56 C, system accuracy was estimated at better than 4% for CWC greater than about 0.1 g/m$^3$. Wind tunnel and other calibrations have established the reliability and high precision of the IKP-2 in high-CWC conditions, and in general support the system accuracy
estimates above (Strapp et al. 2016). Hence, a lower TWC threshold of 0.1g/m3 is taken into account in this study, thereby not using any measured CWC below this threshold.





The 2D-S and the PIP probes record monochromatic 2D shadow images of cloud hydrometeors (ice and/or water) along the flight trajectories. The 2D-S records images of hydrometeors in the size range 10-1280µm at a 10 µm pixel resolution, whereas the PIP records images in the size range 100-6400 µm and beyond (perpendicular to photodiode array and reconstruction of truncated images parallel to the array) at a 100 µm pixel resolution. For both probes, particle size distributions (PSDs) were

produced by image analysis into number concentrations per unit volume of sampled air as a function of their size.

In this paper, the size of ice hydrometeors is given in terms of the maximum diameter $D_{max}$ (e.g. see Leroy et al., 2016) for definition). The size of truncated images and sampling volume are corrected using the method presented in Korolev and Sussman (2000). This reconstruction method allows extrapolating hydrometeor sizes to a maximum size of 2.56 mm for 2D-S and to 12.8 mm for PIP.

In addition, many artefacts can bias PSD estimates from 2D image analysis. Therefore, supplementary post-processing is needed to retain only the natural ice particles. One of the most important causes of artefacts is the shattering of hydrometeors on the tips of OAP probes. During the first HAIC/HIWC field campaign in Darwin, the newest anti-shattering tips were used for 2D-S and PIP in order to reduce the shattering from large ice crystals. In addition, analysis of the time dependent (along the flight trajectory) inter arrival time spectra was performed to determine the cut-off time which separates natural

hydrometeors images from artefact particles (Field et al., 2006; Korolev and Field, 2015; Korolev and Isaac, 2005). It has been shown that both mitigation techniques are needed to maximize the removal of shattering artefacts (Jackson et al., 2014). Furthermore, images of splashed hydrometeors were removed using the ratio between their projected surface area and the surface defined by the box Lx*Ly (e.g. see Dx and Dy in Leroy et al. 2016). Images with an area ratio less than 0.25 were considered as splashed particles and were removed.

Another important correction is related to the sizing out of focus hydrometeors. In our study, the size of out-of-focus particles was corrected using the method presented in Korolev (2007). In addition, noisy pixels (satellite pixels) which may affect hydrometeor images were eliminated in order to get the best estimation of the true $D_{max}$.

Finally, high number concentration of ice particles leads to gaps in the sampling times of OAP probes due to insufficient time to record all hydrometeors images. This probe effect (also called OAP overload or dead time) was taken into account and

would otherwise lead to an underestimation of the number concentration of hydrometeors. While the 2D-S probe overload times are directly registered, the PIP overload is estimated by comparing the number of images in the PIP files to the separately registered total particle counts of particles that passed through the laser beam. The ratio of counted particles (1D information) to recorded particle images (2D information) is used to correct for the concentration. During an OAP overload, 1D counted particles can reach 15000 while only about 10000 have a recorded images, which results an uncertainty by about 50% on the

concentration of hydrometeors (Fontaine, 2014). Further details on post-processing of 2D-S and PIP data are given in Leroy et al. (2016). The individual 2D-S and PIP PSDs were merged into a composite PSD using the algorithm described in Eq. (1). The resolution of the composite PSD is 10 microns (by interpolating the PIP raw PSDs at the 2D-S resolution), and PSDs are averaged over 5s time intervals for improved large particle statistics. The transition zone for changing from 2D-S to PIP data





in the composite spectrum (see equation below) is from a $D_{max}$ of 805 µm (median diameter for a size bin) to 1205 µm. $N(D_{max})$ is given per litre.

$$\sum_{D_{max}=15}^{D_{max}=12845} N(D_{max}) \cdot \Delta D_{max}$$

$$= \sum_{D_{max}=15}^{D_{max}<805} N_{2D-S}(D_{max}) \cdot \Delta D_{max} + C_1(D_{max}) \cdot \sum_{D_{max}=805}^{D_{max}<1205} N_{2D-S}(D_{max}) \cdot \Delta D_{max}$$

$$+ C_2(D_{max}) \sum_{D_{max}=805}^{D_{max}<1205} N_{PIP}(D_{max}) \cdot \Delta D_{max} + \sum_{D_{max}=1205}^{D_{max}=12845} N_{PIP}(D_{max}) \cdot \Delta D_{max} \qquad (1)$$

where $C_1(D_{max}) + C_2(D_{max}) = 1$, with: $C_2(D_{max}) = \frac{D_{max}-805}{1205-805}$.

### 3 Global retrievals of CWC from radar reflectivity simulations

### 3.1 Simulations of radar reflectivity factors: Ze

This section reviews the method used in Fontaine et al., (2014) to retrieve CWC from simulations of radar reflectivity factors ($Ze$), which were applied in that study to cloud data (cloud research radar and OAP images) from tropical MCS formed over the African continent and the Indian Ocean. The data were collected during two aircraft campaigns dedicated to the Megha-Tropiques project (Roca et al., 2015), with no direct measurement of CWC. Their method consists approximated the

backscatter cross section ($Q_{back}$) of natural hydrometeors using with oblate spheroids (Hogan et al., 2011) for ice crystals. In Eq. (2) below, $Ze$ is defined in mm$^6$/m$^3$.

$$Z_e(\overline{As}, f_{ice}(D_{max})) = 1000 \cdot \frac{\lambda^4}{\pi^5 \cdot |K_{w-ref}|} \cdot \sum_{D_{max}=15}^{D_{max}=12845} N(D_{max}) \cdot Q_{back}(\overline{As}, f_{ice}(D_{max})) \cdot \Delta D_{max} \qquad (2)$$

$Q_{back}$ is a function of the axis ratio of the oblate spheroid (here denoted $\overline{As}$, see Eq. (8)) and the ice fraction ($f_{ice}$; Eq. (3)) in the

spheroid. Thereby, $f_{ice}$ is a function of the mass-size relationship (Eq. (4)). Equation (3) also limits the mass of an ice hydrometeor to the mass of an ice sphere of diameter $D_{max}$. $f_{ice}$ also permits the calculation of the dielectric properties of the ice spheroids (Drigeard et al., 2015; Maxwell Garnet, 1904).

$$f_{ice} = min\left(1, \frac{\alpha \cdot D_{max}^{\beta}}{\frac{\pi}{6} \cdot \rho_{ice} \cdot D_{max}^3}\right) \qquad (3)$$





$$m(D_{max}) = \alpha \cdot D_{max}^{\beta} \tag{4}$$

$\overline{As}$ is the average aspect ratio of all hydrometeors and is calculated every 5s according to $N(D_{max})$ and $Z$.

$$\overline{As} = \sum_{D_{max}=15}^{D_{max}=12845} As(D_{max}) \cdot Pi(D_{max}) \tag{5}$$

$As(D_{max})$ for a particle is defined as the ratio of the width (radius perpendicular to $D_{max}$) divided by $D_{max}$. In Fontaine et al. (2014) the calculation of $\overline{As}$ only takes into account all hydrometeors with sizes $D_{max} \leq 2005$ microns which contribute on average to 95% of $Ze$ (Fontaine, 2014). However, processing of 2DS and PIP probes has been further improved since Fontaine et al. (2014; see Leroy et al. 2016) and we decided to take into account all hydrometeors from 15µm to 1.2845cm for $\overline{As}$ calculation. The comparison of the $\overline{As}$ calculation utilized in Fontaine et al. (2014) with the new calculation results in a decrease of $\overline{As}$ of less than 3%, which implicates a negligible impact on the retrieval method. $Pi$ is a weighting function defined to account for the volume occupied by the hydrometeors in the sampled volume:

$$Pi(D_{max}) = \frac{N(D_{max}) \cdot D_{max}^3 \cdot \Delta D_{max}}{\sum_{D_{max}=15}^{D_{max}=12845} N(D_{max}) \cdot D_{max}^3 \cdot \Delta D_{max}} \tag{6}$$

Note that the constant aspect ratio of 0.6 used in Hogan et al. (2011) is rather close to the peak of the $\overline{As}$ frequency distribution presented for African Monsoon MCS and also oceanic MCS over the Indian Ocean (Fontaine et al. (2014)). The average aspect ratio calculated for the HAIC-HIWC dataset is 0.55, which is then very similar to values for various dataset sampled over African continent, United Kingdom, Indian Ocean and North of Australia.

In Fontaine et al. (2014) the exponent $\beta$ of the m(D) power law relationship has been constrained as a function of time from the ice particle images. For this study, no a-priori assumptions on the mass-size relationship of hydrometeors have been chosen and as a consequence a variational approach been applied in order to calculate CWC from $Ze$ reflectivity factor simulations. For a given but variable exponent $\beta_i$ the corresponding pre-factor $\alpha_i$ is calculated in order to match the measured reflectivity Z with the simulated Ze. $\beta_i$ is varying stepwise between 1 to 3 (by increments of 0.01). Thereby, the range of potential solutions are explored using the oblate spheroids approximation with the T-matrix method (Mishchenko et al., 1996) calculating $Q_{back}$ of each spheroid. Hence, for a given 5s data point, 201 calculations of $\alpha_i$ and 201 calculations of corresponding $CWC(\beta_i)$ are performed (see Eq. (10)):

$$CWC(\beta_i) = 10^3 \cdot \sum_{D_{max}=15}^{D_{max}=12845} N(D_{max}) \cdot \alpha \cdot D_{max}^{\beta} \cdot \Delta D_{max} \tag{7}$$

For each 5s data point, from the 201 possible $CWC(\beta_i)$ values, an average value $\overline{CWC}$ is deduced (Eq. (11)):

$$\overline{CWC} = \frac{1}{N_{tot}} \cdot \sum_{\beta_i=1}^{\beta_i=3} CWC(\beta_i) , \tag{8}$$


where $N_{tot} \leq 201$, since the minimum value allowed for $\alpha_i$ is the mass of an empty sphere (air density).

Figure 1 shows two examples (flight 9 and 12) with all possible $CWC(\beta_i)$ retrievals (colour band), average $\overline{CWC}$ (red line), and $CWC_{IKP}$ measured by the IKP (overlaid black line). The example in Figure 1-a) shows results from flight 9, where the F-20 research aircraft flew in the more stratiform part of the cloud system (w~0m/s), whereas results from flight 12 shown in Figure 1-b) represents a case with more signatures of convective updrafts. Overall, Figure 1 demonstrates that the variational retrieval method produces a large variability of possible $CWC(\beta_i)$ for each 5s data point. In general, the average $\overline{CWC}$ (red line) is close to $CWC_{IKP}$. The bandwidth of all possible solutions $CWC(\beta_i)$ as a function of time is calculated from the difference $\Delta CWC=\max(CWC(\beta_i))–\min(CWC(\beta_i))$ between the maximum and the minimum values of $CWC(\beta_i)$. On average it is found that $\Delta CWC$ accounts for 61% of $\overline{CWC}$ (median relative error is 64%) for the entire dataset. Finally, the calculations reveal that 80% of $CWC(\beta_i)$ satisfy the condition $CWC_{IKP} = \overline{CWC} \pm 32\%$, where no a priori assumptions on mass-size relationships were applied and $\beta_i$ linearly varies between 1 and 3, thereby producing equally eligible solutions $CWC(\beta_i)$.

In general, and for each given 5s data point, maximum CWC is obtained for $\beta=1$ and minimum CWC for $\beta=3$. For $\beta=1$, ice hydrometeors below $D_{max} = 200\mu m$ (sometimes even below 300 µm) may reach the maximum density of $0.917g/cm^3$, while for $\beta=3$ the density of oblate spheroids is constant as a function of $D_{max}$ (see Eq. (3)); where the density of icy spheroids is equal to $0.917*f_{ice}$). The impact of $\beta$ on $Ze$ and the retrieved CWCs is illustrated in Figure 2, respectively. These Figs. show for different $\beta$ (=1, 2 and 3) values, the cumulative sum of $Ze$ as a function of $D_{max}$ normalized by the measured radar reflectivity factor $Z$ (in dBZ; Figure 2-a.), and the cumulative mass of ice crystals as a function of $D_{max}$ normalized by $\overline{CWC}$ (Figure 2-b). Obviously, the normalized cumulative sum of Ze is always equal to 1, when integrating up to largest hydrometeors, independently of $\beta$, while this is not the case for the normalised cumulative mass of ice crystals. For $\beta=1$, Ze is reached sooner ($D_{max} \approx 1700\mu m$) than for $\beta=3$ ($D_{max} \approx 3000\mu m$). The likely explanation is that with increasing $\beta$, the backscattered energy is increased for large hydrometeors and the mass contribution of smaller hydrometeors is considerably reduced since the contribution of numerous smaller hydrometeors (compared to larger hydrometeors) on retrieved CWC is decreasing with increasing $\beta$.

## 3.2 CWC deviations from T-matrix simulations of reflectivity with respect to IKP direct measurements

This section focuses on the potential error in CWC retrievals from T-Matrix simulations of radar reflectivity factors (at frequency of 94GHz) for populations of ice hydrometeors approximated with oblate spheroids. Therefore, the relative errors (in per cent) of retrieved $\overline{CWC}$ with respect to reference $CWC_{IKP}$ (measured by the IKP-2 probe) are calculated and then analysed as a function of microphysical properties of ice hydrometeors such as total concentration (($N_T$); Fig. 3), temperature $T$ (Fig. 4), maximum size of hydrometeors in PSDs ($max(D_{max})$; Fig. 5)), total cloud water content $CWC_{IKP}$ (Fig. 6) and radar reflectivity $Z$ (Fig. 7). Blue lines in Figure 3-7 (upper charts) display median trends obtained when the relative errors of $\overline{CWC}$ are plotted as a function of the crystal number concentration $N_T$, the temperature T, the maximum encountered crystal size





$max(D_{max})$, the $CWC_{IKP}$, and the radar reflectivity $Z$. Bottom and top whiskers of the error bars represent the 25th and 75th percentiles of the relative error of $\overline{CWC}$ (with respect to $CWC_{IKP}$). Lower charts in Figs. 3-7 illustrate the number of samples used for the calculation of the respective data points in discrete intervals of $N_T$, T, $max(D_{max})$, $CWC_{IKP}$, and $Z$.

From Fig. 3 (blue line) it appears that $\overline{CWC}$ resulting from T-Matrix simulations approximating ice hydrometeors with oblate

spheroids, are poorer at the lower ($N_T$<100/L) and higher ($N_T$>5000/L) ranges of number concentrations. In particular, the reference $CWC_{IKP}$ is underestimated for small $N_T$ and overestimated for larger $N_T$. Furthermore, Fig. 4 (blue line) seems to illustrate that this method increasingly overestimates with decreasing temperature (blue line in Figure 4). For example, $\overline{CWC}$ exceeds $CWC_{IKP}$ at 220 K (±5 K) by about 25%. Finally, the relative errors of $\overline{CWC}$ with respect to $CWC_{IKP}$ slightly but continuously increase with the maximum size of hydrometeors within the respective data point (blue line in Figure 5), where

the relative error of $\overline{CWC}$ reaches +25 % when $max(D_{max})$ = 1 cm, versus ~0% for $max(D_{max})$ = 800 µm.

**3.3 Correction functions for CWC retrievals from T-matrix simulations of reflectivity**

As a consequence of the above findings, three different types of corrections are studied, in order to first quantify the limitations of the oblate spheroid approximation and second to suggest suitable correction functions that use in-situ measured quantities over the entire dataset with $CWC_{IKP}$ larger than 0.1g/m³. The impact of these corrections on relative errors as a function of $N_T$,

15    T, and $max(D_{max})$ are added to Figure 6.

- Red lines in Figs. 3-7 represent the relative error of $\overline{CWC}$ * f($N_T$) after applying a correction function $f(N_T)$ as a function of $N_T$ with:

$$f(N_T) = 0.84 \cdot (-0.3012 \cdot log_{10}(N_T)^3 + 2.658 \cdot log_{10}(N_T)^2 - 7.758 \cdot log_{10}(N_T) + 8.493) \qquad (9)$$

- Grey lines in Figs. 3-7 represent the relative error of $\overline{CWC}$ * f($T$) after applying a correction function f($T$) as a function

of $T$ with:

$$f(T) = 0.84 \cdot (0.006528 \cdot T - 0.517) \qquad (10)$$

- Black lines in Figs. 3-7 represent the relative error of $\overline{CWC}$ * f($max(D_{max})$) after applying a correction function f($max(D_{max})$) as a function of $max(D_{max})$ with:

$$f(\max(D_{max})) = 0.84 \cdot (2.092.\,10^{-9} \cdot \max(D_{max})^2 - 3.869.\,10^{-5} \max(D_{max}) + 1.15) \qquad (11)$$

Without the above correction functions the retrieved initial $\overline{CWC}$ 5s averages are larger than $CWC_{IKP}$ by about 19% on average (with a median value of +16% (Table 1, first row)). Therefore, all three correction functions (Eq. (9) to (11)) have in common a median factor of 0.84 reducing the initial $\overline{CWC}$ such that $\overline{CWC}$ *f(X; X=$N_T$, T, $max(D_{max})$) better matches $CWC_{IKP}$. The expressions in brackets of Eq (9) to (11) try to redistribute the relative error in $\overline{CWC}$ from T-Matrix simulations over the entire range of observed $N_T$, T, and $max(D_{max})$ values, but have negligible impact on the median relative error itself. Even though no

correction functions for $\overline{CWC}$ have been proposed as a function of $CWC_{IKP}$ and $Z$, Figs. 6 and 7 illustrate also the impact of



correction functions $N_T$, $T$, and $max(D_{max})$ (Eq. (9), (10) and 11)) on the redistribution of the relative error also as a function of $CWC_{IKP}$ and $Z$.

Figure 3 reveals that $f(N_T)$ (Eq. (9)) decreases biases of retrieved $\overline{CWC} * f(N_T)$ over the entire $N_T$ bandwidth, while $f(T)$ (Eq. (10)) and $f(max(D_{max}))$ (Eq. (11)) do not change the shapes of the relative error lines as compared to uncorrected $\overline{CWC}$ relative errors (Fig. 3). Also the function $f(N_T)$ (Eq. (9)) also generally improves the retrieval relative errors $\overline{CWC} * f(N_T)$ when plotted as a function of $T$ (Fig. 4) and $max(D_{max})$ (Fig. 6). Furthermore, the $f(T)$ correction function (Eq. (10)) reduces the differences between $\overline{CWC} * f(T)$ and $CWC_{IKP}$ as a function of $T$ (Figure 4). However, this correction function performs less well as a function of $N_T$ (Fig. 3, grey line) and as a function of $Z$ (Fig. 7, grey line). Finally, the $f(max(D_{max}))$ correction function (Eq. (11)) reduces the relative error of retrieved $\overline{CWC} * f(max(D_{max}))$ as a function of $max(Dmax)$ in Figure 5, but does not have much impact on the the shape of the relative error distributions as a function of $N_T$, $T$, $CWC_{IKP}$, and $Z$ (Figs. 3, 4, 5, 7) relative to uncorrected $\overline{CWC}$ relative errors.

In addition, rows 2-4 of Table 1 present mean (average), median, 10th, 25th, 75th, and 90th percentiles of the relative error after applying the correction functions, with an obvious decrease of mean and median values and corresponding shift of relative error distribution percentiles.

Overall, the $f(Nt)$ correction seems most efficient to remove the CWC bias. Heymsfield et al., (2013) showed that total concentrations of ice hydrometeors tend to increase with decreasing temperature in tropical MCS for temperatures -60°C<T<0°C. This evolution of the increasing total concentration of hydrometeors related to decreasing temperatures is therefore suggested as the key to explaining trends in relative CWC errors as a function of $N_T$ and $T$.

Figure 8 summarizes the above results by showing probability distribution functions (PDF) of $\overline{CWC} * f(X; X=N_T, T, max(D_{max}))$ versus $CWC_{IKP}$. Imperfections of the corrections described by Eq. (10) and (11) are clearly visible in Fig. 8-c and Fig. 8-d, where high $CWC_{IKP}$ values are still overestimated (as in Fig. 8-a)) by $\overline{CWC} * f(T)$ and $\overline{CWC} * f(max(D_{max}))$, respectively. The correction function $f(N_T)$ (Eq. (9)) produces the best results (Figure 8-b), where the maximum of the PDF in $\overline{CWC} * f(N_T)$ versus $CWC_{IKP}$ representation follows the line y=x.

## 4 Uncertainties in ice particle concentrations and impact on results

This section investigates the impact on above findings (retrieved CWC) of uncertainties in crystal concentrations, in particular taking into account a possible shattering effect. As discussed in section 3.2 the relative errors increase with total number concentrations, showing that retrieved CWCs overestimate $CWC_{IKP}$ by about 50% for very large concentrations of hydrometeors which can reach $10^4$ hydrometeors per liter in most convective parts of sampled data (Figure 3-a). In order to investigate the impact of uncertainties in number concentrations on the retrieved CWC we apply two types of functions on the measured PSDs, thereby increasing number concentrations in PSD.

First, a function $f_{shatt}$ is applied to the PSD in order to increase concentrations of hydrometeors in the first PSD size bin (5-15 µm) by about 50%, while concentrations of hydrometeors larger than 500 microns remained unchanged. The function $f_{shatt}$





decreases in a logarithmic way with $D_{max}$ from first bin to 500µm. $f_{shatt}$ is applied to PSD such that $N'(D_{max})=f_{shatt}(D_{max})*N(D_{max})$ and aims producing new PSDs where the optimized probe tips still would have produced shattered crystal fragments and/or removal processing would have failed to remove numerous shattered ice particles. Then, the retrieval method (see section 3.1) is applied to these new PSDs in order to calculate new values for $\alpha_i$ and subsequently $CWC_i$ (hereafter $\alpha_{i,fshatt}$ and $CWC_{i,fshatt}$).

For the purpose of this section 4, the method was only applied for $\beta = [1, 2, 3]$ in order to get a good idea of the maximum impact of possible shattering artefacts. Results are presented in terms of relative errors in Table 2 for $\alpha_{i,fshatt}$ and Table 3 for $CWC_{i,fshatt}$, respectively. Relative errors in (%) are calculated with respect to coefficients $\alpha_i$ and $CWC_i$ (for $\beta=[1, 2, 3]$) calculated in section 3 for non-modified original $N(D_{max})$ size distributions and without correction functions discussed in section 3.3. (Eq. (9) to (11)). The relative errors illustrate that the chosen concentration increase of solely small hydrometeor

sizes has very limited impact on retrieved $\alpha_i$ and $CWC_i$. Indeed, we observe that median relative error of the prefactor $\alpha_{i,fshatt}$ with respect to $\alpha_i$ is roughly -3% for $\beta=1$ and 0% for $\beta=2$ and $\beta=3$. 1[th] and 99[th] percentiles are given in order to demonstrate that relative errors in $\alpha_{i,fshatt}$ are small over the entire data set. Consecutively, median relative errors of $CWC_{i,fshatt}$ with respect to $CWC_i$, are of the order of 4%, 3%, and 1% for $\beta=1$, 2, and 3, respectively. The 99[th] percentile of the relative error does not exceed 10% in retrieved CWC.

Second, a simple concentration uncertainty factor of 1.5 is applied over the entire measured size range, which increases the number concentration by 50% such that $N'(D_{max})= (1.5*N(D_{max}))$. Note that 50% is approximatively the missed number of ice crystal images by the PIP probe, due to the overload in high concentration of ice crystals, though data have been corrected for overload times (see section 2 and Fontaine, (2014)). Again, simulations of the reflectivity factor with modified $N'(D_{max})$ *size* were performed with resulting prefactor $\alpha_{i,50\%}$ and $CWC_{i,50\%}$. Results of the comparison of $\alpha_{i,50\%}$ and $CWC_{i,50\%}$ with $\alpha_i$ and

$CWC_i$, respectively, are also presented in Table 2 and Table 3. Globally, this second scenario of concentration enhancement of original PSD has a larger impact on retrieved prefactor ($\alpha_i$) and $CWC_i$, than has been the case for the first scenario. Indeed, adding 50% to the concentrations of hydrometeors results in a median decrease of prefactor $\alpha_{i,50\%}$ with respect to $\alpha_i$ of -18% for $\beta=1$ and -20% for $\beta=2$ *and 3*. The forced decrease in $\alpha$ (from $\alpha_{i,50\%}$ to $\alpha_i$) goes along with an increase of $CWC_{i,50\%}$ with respect to $CWC_i$ by about +29% (for $\beta=1$) and +27% for $\beta=2$ and 3. Indeed for a given radar reflectivity factor we simulate

(and measure!) the same $Ze$ (or $Z$) for $N'(Dmax)$ and $N(D_{max})$ with decreasing $\alpha$ ($\alpha_{i,50\%}$ compared to $\alpha_i$), while the CWC is increasing by almost 30% ($CWC_{i,50\%}$ case compared to $CWC_i$). Hence, if two different size distributions produce an identical Ze, CWC can be significantly different. In other words, for a same CWC associated to two underlying different size distributions Ze may differ significantly, even without taking into account uncertainties in concentration measurements.

## 5 Discussion and Conclusions

The presented study is based on simulating radar reflectivity factors $Ze$ such that the simulations match the measured radar reflectivity factors at 94 GHz. Since mass-size relationships are considered to be unknown, the $Ze$ simulations explore a wide range of possible solutions of ($\alpha_i$, $\beta_i$), varying $\beta$ in the range of 1 to 3. This produces a series of possible CWCs for a given





data point, one for each value of $\beta$. From this series, the average value $\overline{CWC}$ over all values of $\beta$ is calculated. On average it is found that the difference $CWC(\beta=1)-CWC(\beta=3)$ is approximately 64% of the average $\overline{CWC}$. 77% of Darwin data points meet the condition of $CWC(\beta=3) \leq CWC_{IKP} \leq CWC(\beta=1)$, which goes along with the relation $CWC_{IKP} = \overline{CWC} \pm 32\%$. However, the retrieved $\overline{CWC}$ values are generally larger than $CWC_{IKP}$ by about 16% (median value), which illustrates that the

5 approximation of ice oblate spheroids tends to underestimate the backscattered energy of real reflectivity measurements of ice hydrometeors at 94GHz, and a constant factor of 0.84 could be applied as a first order correction of retrieved $\overline{CWC}$.

One of the possible explanations is that the calculation of the average aspect ratio from 2D images (Eq. (5) and (6)) which has been adopted as axis ratios for the approximation ice oblate spheroids may be somewhat too large. A way to investigate impact and calculation of axis ratio for ice oblate spheroids approximations would be to use $As(D_{max})$ instead of $\overline{As}$ (Eq. (5)) in

simulations of radar reflectivity factors.

This study also demonstrated that the total concentration of ice hydrometeors could be used as input for a correction algorithm that minimizes differences between $\overline{CWC}$ and $CWC_{IKP}$, and that this parameter was the best of several parameters evaluated for this purpose. These differences, before correction, were found to increase with increasing ice concentration, with $\overline{CWC}$ underestimating $CWC_{IKP}$ at low ice concentrations, and overestimating $CWC_{IKP}$ at high concentrations.

Another attempt of explanation is the uncertainty of measured total crystal concentrations which could partly explain large relative errors at high concentrations. Indeed, for very high concentrations we find a higher relative error of about +50% as compared to IKP measurements. However, according to the results of section 4 of this study, an uncertainty of 50% in ice crystals concentrations, can solely explain 30% of the relative errors. Likewise, concentration errors related to large crystal shattering, could not explain more than 35% of the relative errors at very high concentrations of ice hydrometeors. However,

this cannot explain the negative relative errors for low ice crystal concentrations. Also at very low ice crystals concentrations, oblate spheroids approximation of crystals could be not sufficiently adapted, since for low concentrations real shapes of ice hydrometeors might be even more important due to the lack of any averaging process over all possible shapes and possible orientations as should be more likely the case for higher concentrations.

**Acknowledgments**

The research leading to these results has received funding from (i) the European Union's Seventh Framework Program in research, technological development and demonstration under grant agreement n°ACP2-GA-2012-314314, (ii) the European Aviation Safety Agency (EASA) Research Program under service contract n° EASA.2013.FC27, and (iii) the Federal Aviation Administration (FAA), Aviation Research Division, and Aviation Weather Division, under agreement CON-I-1301 with the Centre National de la Recherche Scientifique. Funding to support the Darwin flight project was also provided by the NASA

Aviation Safety Program, the Boeing Co., and Transport Canada. Additional support was also provided by Airbus SAS Operations, Science Engineering Associates, the Bureau of Meteorology, Environment Canada, the National Research Council





of Canada and Universities of Utah and Illinois. The authors thank the SAFIRE facility for the scientific airborne operations. SAFIRE (http://www.safire.fr) is a joint facility of CNRS, Météo-France and CNES dedicated to flying research aircraft.

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





**Table 1: Relative errors of calculated simulation CWCs relative to CWC$_{IKP}$, in %**

| Relative error | mean | 10th | 25th | 50th | 75th | 90th |
|---|---|---|---|---|---|---|
| $\dfrac{\overline{CWC} - CWC_{IKP}}{CWC_{IKP}} \cdot 100\%$ | 19 | -14 | 2 | 16 | 32 | 54 |
| $\dfrac{\overline{CWC} \cdot f(N_T) - CWC_{IKP}}{CWC_{IKP}} \cdot 100\%$ | 2 | -24 | -12 | -1 | 12 | 28 |
| $\dfrac{\overline{CWC} \cdot f(T) - CWC_{IKP}}{CWC_{IKP}} \cdot 100\%$ | 4 | -24 | -11 | 1 | 15 | 34 |
| $\dfrac{\overline{CWC} \cdot f(\max(D_{max})) - CWC_{IKP}}{CWC_{IKP}} \cdot 100\%$ | 3 | -26 | -12 | 0 | 14 | 33 |

5     **Table 2: Relative errors of retrieved prefactor α$_{i,fshatt}$ and α$_{i,50\%}$ with respect to α$_i$ in % for β= [1, 2, 3].**

|  | $(\alpha_{i,fshatt} - \alpha_i)/\alpha_i$ | | | $(\alpha_{i,50\%} - \alpha_i)/\alpha_i$ | | |
|---|---|---|---|---|---|---|
|  | 1th | median | 99th | 1th | median | 99th |
| β=1 | -3% | -3% | 1% | -21% | -18% | -18% |
| β=2 | -3% | 0% | 0% | -20% | -20% | -18% |
| β=3 | -3% | 0% | 0% | -24% | -20% | -18% |





5    **Table 3: Relative errors of retrieved $CWC_{i,fshatt}$ and $CWC_{i,50\%}$ with respect to $CWC_i$ in % for β= [1, 2, 3].**

| | $(CWC_{i,fshatt}-CWC_i)/CWC_i$ | | | $(CWC_{i,50\%}-CWC_i)/CWC_i$ | | |
|---|---|---|---|---|---|---|
| | $1^{th}$ | median | $99^{th}$ | $1^{th}$ | median | $99^{th}$ |
| β=1 | -1% | +4% | +9% | +23% | +29% | +32% |
| β=2 | --1% | +3% | +9% | +24% | +27% | +30% |
| β=3 | +1% | +1% | +5% | +22%% | +27% | +30% |





Figure 1: Condensed water content CWC as a function of time for two HAIC-HIWC flights during the Darwin 2014 flight campaign. In black $CWC_{IKP}$, in red the average $\overline{CWC}$ deduced from all possible simulations (varying β and constraining α) of the measured $Z$. Blue to green colored band shows $CWC(\beta_i)$ calculations when β varies from 1 (blue) to 3 (green). a) Results for flight 9 and b) Results for flight 12.







**Figure 2: a) Cumulative sum of simulated Ze over measured total Z as a function of $D_{max}$. b) Cumulative sum of ice crystal mass for constant β over the $\overline{CWC}$ averaged over all ($α_i$, $β_i$) solutions represented as a function of $D_{max}$. Full lines represent median and dashed lines 25th and 75th percentiles. Blue lines for $β_i$ =1, black lines for $β_i$=2, and red lines for $β_i$=3.**

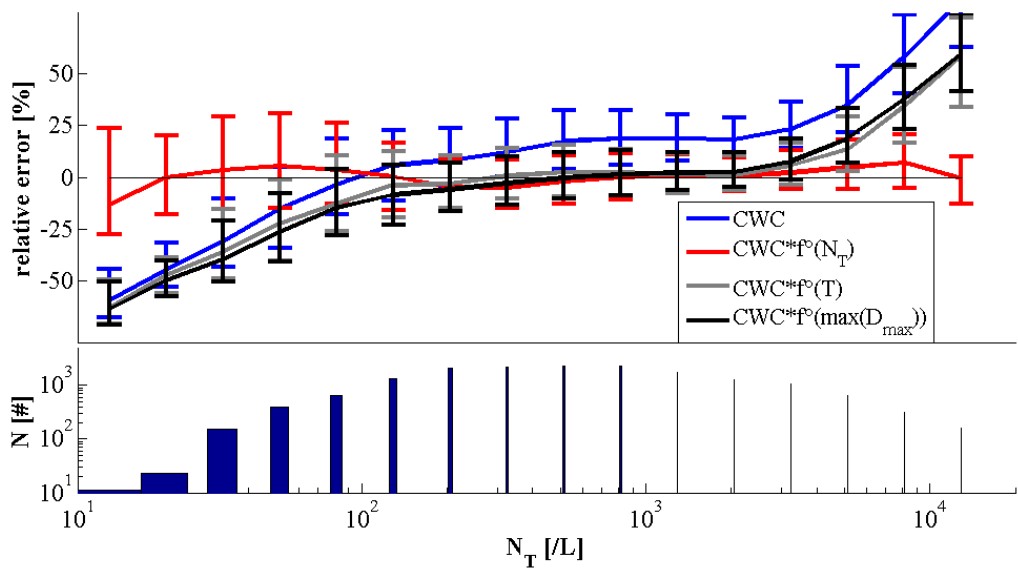

**Figure 3: a) Relative errors of $\overline{CWC}$ s with respect to $CWC_{IKP}$ (as defined in Table 1) as a function of total PSD number concentration $N_T$. The errors are presented with and without the three suggested correction functions for $\overline{CWC}$. b) number of 5s data points used for the statistics on y-axis as a function of $N_T$ intervals.**





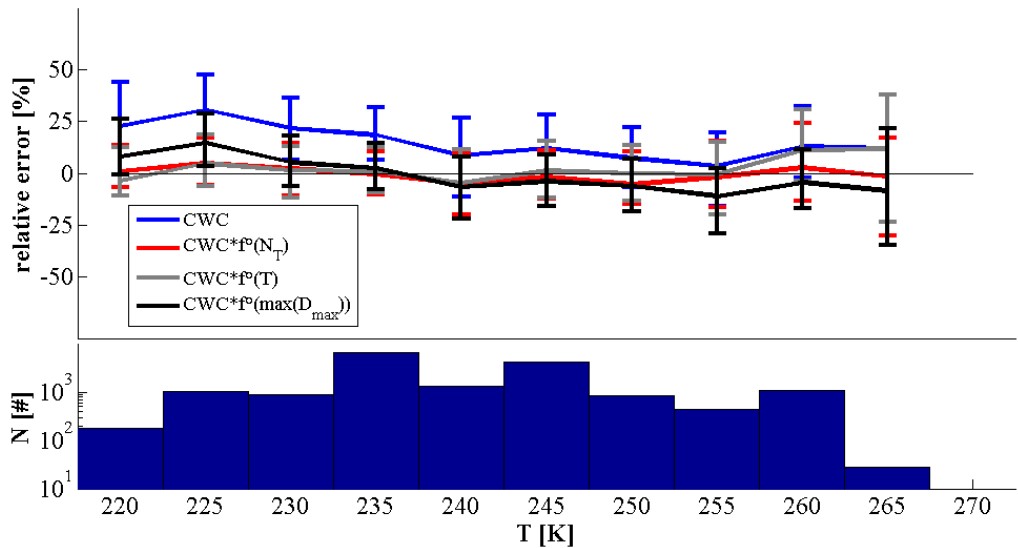

**Figure 4: Same as Figure 3, but represented as a function of temperature T on the x-axis.**

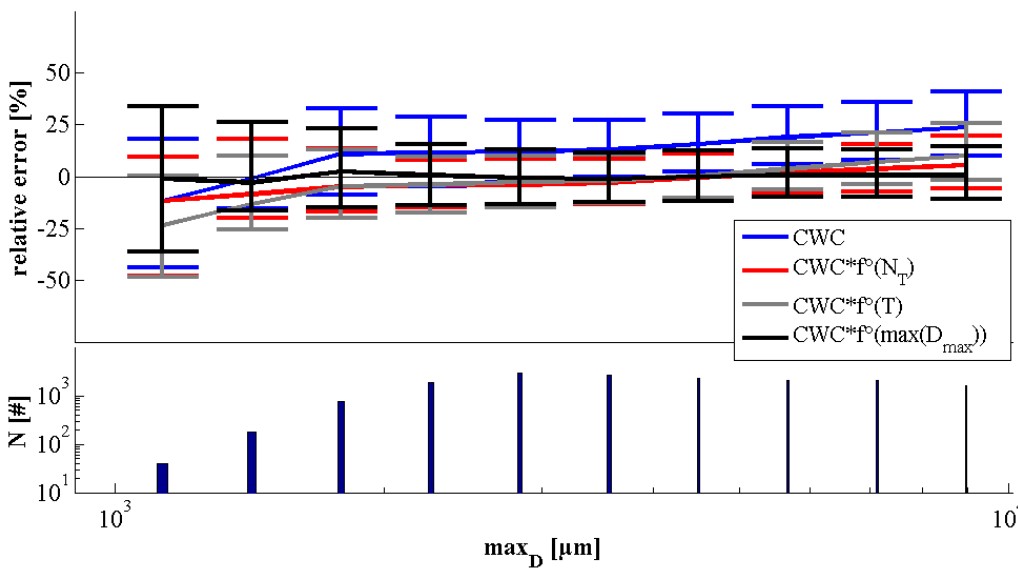

5   **Figure 5: Same as Figure 3, but represented as a function of the maximum size of hydrometeors in the PSD** *max($D_{max}$)*.



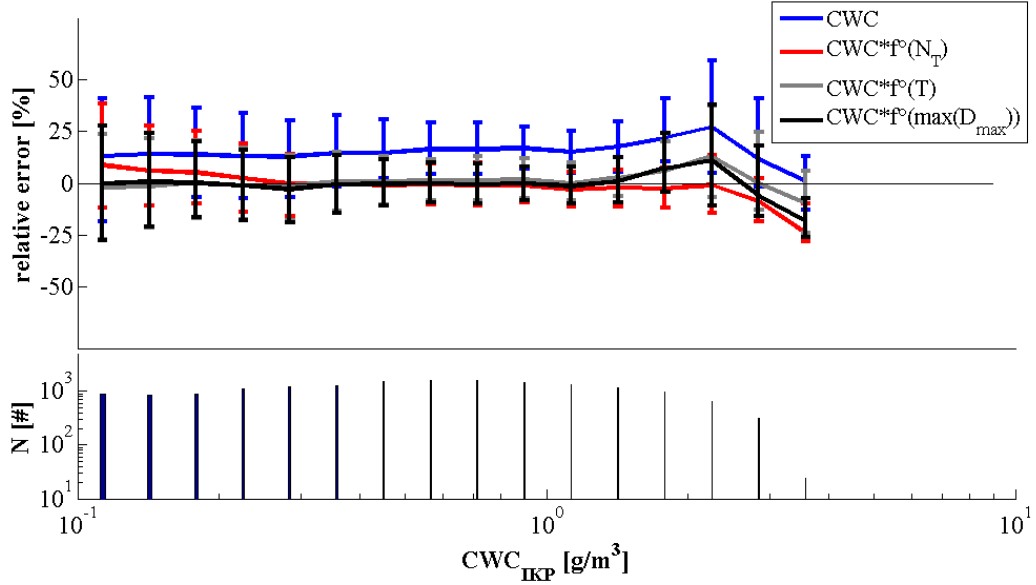

**Figure 6: Same as Figure 3, but represented as a function of CWC_IKP.**

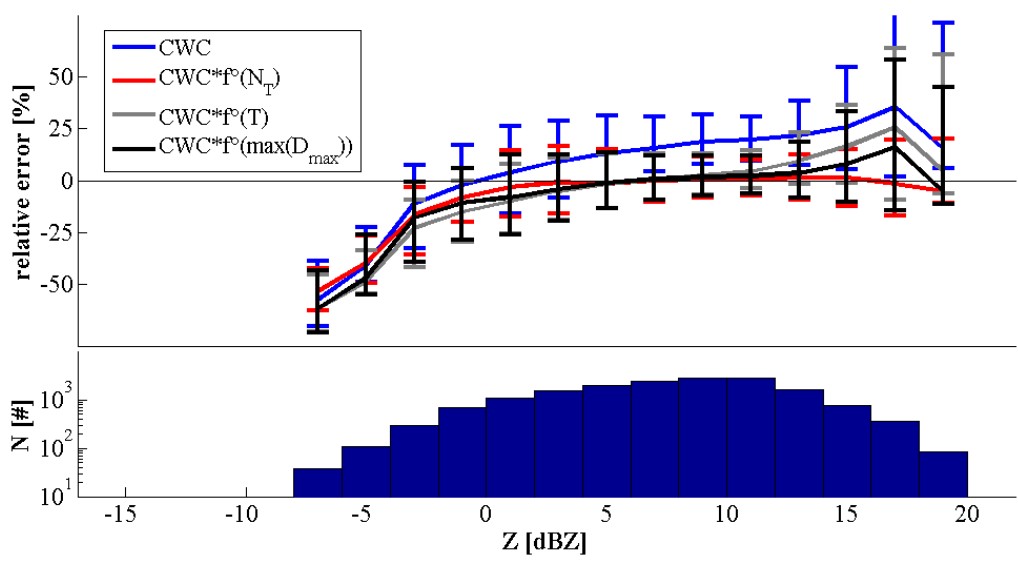

5 **Figure 7: Same as Figure 3, but represented as a function of radar reflectivity Z.**





**Figure 8: Probability distribution functions of $\overline{CWC}$ on y-axis calculated as a function of $CWC_{IKP}$ on x-axis. Probabilities are represented by the color scale and were normalized by the number of data points. a) No correction is applied to average $\overline{CWC}$. b) Correction f($N_T$) described by Eq. (9) is applied to $\overline{CWC}$. c) Correction f($T$) described by Eq. (10) is applied to $\overline{CWC}$. d) Correction f($max(D_{max})$) described by Eq. (11) is applied to $\overline{CWC}$.**