# Peer review of "Radar Reflectivity Factors Simulations of Ice Crystal Populations from In-Situ Observations for the Retrieval of Condensed Water Content in Tropical Mesoscale Convective Systems"

_Atmospheric Measurement Techniques, 2016_

## Author Comment (AC1) · 25 Oct 2016

The following reference is missing (Strapp et al. 2016): Strapp, J. W., G. A. Isaac. A. Korolev, T. Ratvasky, R. Potts, P. May, A. Protat, P. Minnis, A. Ackerman, A. Fridlind, J. Haggerty, and J. Riley, 2016: The High Ice Water Content (HIWC) Study of deep convective clouds: Science and technical plan. FAA Rep. DOT/FAA/TC-14/31, available at http://www.tc.faa.gov/its/worldpac/techrpt/tc14-31.pdf. 105 pps.

---

## Referee Comment (RC1) · Anonymous Referee #1 · 16 Dec 2016

Manuscript number: AMT 2016-336

Review of: "Radar Reflectivity Factors Simulations of Ice Crystal Populations from In-Situ Observations for the Retrieval of Condensed Water Content in Tropical Mesoscale Convective System"

Authors: E. Fontaine, D. Leroy, A. Schwarzenboeck, J. Delanoë, A. Protat, F. Dezitter, A. Grandin, J. W. Strapp, L. E. Lilie

General comments: Fontaine et al. (2014) parameterized ice mass-diameter relationships and showed the impact on the Condensed Water Content (CWC) from reflectivity field of anvil clouds. They employed a combined analysis of particle from 2-D-array probes and associated reflectivity from cloud radar. They considered ice crystals as oblate spheroids in the T-Matrix calculations. In this manuscript, they proposed a correction function for CWC retrieval from T Matrix simulation of reflectivity. The manuscript evaluates CWC retrieval method from radar reflectivity factors simulations and in-situ airborne observations within ice crystal regions of tropical mesoscale convective systems. The proposed method is based on the combination of multiple observational datasets (optical array probes [PIP, 2D-S], isokinetic probe [IKP, IKP-2], research cloud radar) and simulations of radar reflectivity factors using a T-matrix approach. After a description of the HAIC-HIWC field campaign dataset and associated processing, the authors present briefly an evaluation based on an ensemble of mass-size relationships to retrieve cloud radar reflectivity factors by assuming ice crystals to be oblate, so that it might match with reflectivity measurements as discussed in Fontaine et al. (2014). Overall this method seems to underestimate radar reflectivity observations. The authors investigates after different solutions to limit oblate spheroid assumption by exploring three different functions driven by total PSD number concentration, temperature and the maximum size of hydrometeors. The best solution of this investigation relies on the total number concentration of ice crystals. Finally, an exploration of the uncertainties due to direct ice measurements with shattering effect is performed, showing that it could partially (but not entirely) account for underestimations. It is not clear the cloud type the authors are discussing, there is no details about the aircraft mission and the specific type of flights. This is very important because it limits the range of applications of these specific relationships; it is only for Cirrus, convective out flow? The title says Tropical mesoscale convective systems, so it includes the core of convective cells, where I doubt this study is valid. The authors exploit quite well the potential of their datasets and T-matrix simulations as complementary of the Fontaine et al. (2014) paper. They used the IKP-2 to test and evaluate the results obtained in his 2014 paper. It is why, in my opinion, somewhere in the title, it should include the word "evaluation". I also, recommend improving the discussion and high-

light the topics about the microphysics of the clouds they are studying, the applications and the contributions of their new results (what is the manuscript contribution after Fountaine et al. 2014). Results presented within this manuscript should be confronted with more literature background and associated comparative discussion about others CWC retrieval methods and implications expected for the community. In addition, some grammatical/spelling errors make, sometimes, difficult to understand some sentences. The authors could sometimes refer to Fontaine et al. (2014) paper or merge different sets of equations to be clearer. Once all of these concerns are addressed, I believe the paper will make an important contribution to the literature.

Specific comments:

Page 1, abstract: please add a comment for the section 4 about uncertainties due to measurements.

Page 2, lines 1-5: please develop. It is somewhat dropped from heaven.

Page 2, line 11: "...than Mie solutions can be applied...". Please explain for which wavelength.

Page 2, line 13: "...difficult problem of ice crystals".

Page 2, line 16: "...recognition techniques, even..."

Page 2, line 17: "...Cloud Particle Imager [2.3$\mu$m resolution, e.g. Mioche, 2010]". Also Page 13, line 23 in the references section: be more precise about the study of Mioche. Is it a Master thesis ? PhD thesis ? ...?

Page 2, line 21: Locatelli and Hobbs 1974 is missing in the references section.

Page 2, line 22: Please replace "Even though..." by "Despite the fact that..."

Page 2, line 23: "...has been utilized, the study of Hogan et al. (2011)..."

Page 2, line 25: There are two Fontaine et al. 2014 in the references section. Please

use 2014a and 2014b. Also for 2014a define in the references section if this work is a master thesis or a PhD thesis.

Page 2, line 26: The acronym of condensed water content is already define in the abstract. Please remove "(CWC)".

Page 2, line 26: "and Drigeard et al. (2015) used their results to simulate…"

Page 2, line 29: "…Dezitter et al. 2013…". The date of the paper is missing in the references section.

Page 3, line 4: "…simulation method (Fontaine et al., 2014) also used in this study is reviewed".

Page 3, line 17: "…the 2D-Stereo probe (2D-S)…"

Page 3, line 18: Please remove "…(DMT)". Not used after.

Page 3, line 19: "The Isokinetic Evaporator Probe (IKP-2: Davison et al., 2016)…"

Page 3, line 30: "overestimates (?) (Strapp et al., 2016)"

Page 3, line 30: "Hence, a lower total water content threshold…"

Page 4, line 15: Please write in the chronological order.

Page 4, line 18: "Images with an area ration lower than 0.25…". It is not clear where the value 0.25 comes from, as well Lx and Ly. Please explain.

Page 4, line 21: Korolev (2007) is missing in the references section.

Page 4, line 21: noise pixel (satellite pixel) – please explain

Page 5, line 12: "…in that study to RASTA data and OAP images from tropical…"

Page, line 15: "Megha-Tropiques project (Roca et al., 2015),…"

Page 5, Equation 2: Please define $\lambda$ and Kw-ref

Page 5, line 22: Please write in the chronological order.

Page 6, line 7: "average to 95% of Ze. However…"

Page 6, lines 7-8: "However, processing of 2D-S and PIP probes has been further improved by Leroy et al. (2016) and …"

Page 6, line 22: "…between 1 to 3 by increments of 0.01. Thereby…"

From page 5 line 10 to page 7 line 1: it is very hard to read and understand this section. Please rewrite this part. Information about equation arrive after explanations, sometimes you missed to define each member of equations, and generally the sentences are really poor in terms of English. To be more readable, please present all equation one after one. And then explain everything.

Page 7, line 9: "…for 61% of CWC (with 64% for the relative error)…"

Page 7, line 9: "…for the entire dataset." What is the entire dataset for you ? Only flights 9 and 12 ? Or more ? Please explain.

Page 7, line 14: "…(see Eq. 4);…" ?

Page 7, line 15: "The impact of…" Please return to the line.

Page 7, line 15: "…is illustrated in Figure 2. One can notice that for different $\beta$ values…"

Page 7, line 18-19: In the same sentence you used "Obviously, when, while"…Come on!!

Page 7, line 28: Therefore the relative errors of retrieve CWC…"

Page 8, line 12: "…in order to i) quantify the limitations of […] and ii) suggest suitable…"

Page 8, line 15: "…are added to Figures 3-8." And please chose between Fig or Figure / Equation or Eq. in the text and remain coherent.

Page 8, equations 9-11: Where these equations come from? Please develop!

Page 8, line 25: "...retrieved initial CWC 5s averages..."

Page 9, line 1: "...(Eqs. 9-11)..."

Page 9, line 7: "However this correction function performs less well..." Please rewrite!

Page 9, line 19: Please remove the acronym "PDF". You used it only twice in 5 lines...

Pages 7-9, sections 3.2 and 3.3: In my opinion sections 3.2 and 3.3 should be merged. It is very frustrating and not so informative to discuss only about one curve and have explanations of the others in the following section.

Page 9, line 30: "...measured PSDs."

Page 10, line 15: "Second, a simple..." Where is the "first" ? If you would like to use "second" please use "first" somewhere before. Otherwise use other suitable link-word to argue.

Page 10, line 18: Remove "Again". "Simulations of the reflectivity..."

Page 10, line 23: "...for both $\beta$=2 and 3."

Page 10, line 25: "(resp. measure) the same Ze (resp. Z)..."

Page 11, line 8: "...ice oblate spheroids might be somewhat..."

Pages 10-11: In my opinion this section looks like rather as a conclusion than a discussion. Please develop also the discussion part.

References section:

Please write in the chronological and alphabetical order (e.g Korolev).

Brown and Francis 1995 is mentioning twice p12 lines 4-5 and p12 lines 17-18. I recommend to use the second one that seems to be more up to date.

[Figure]

Tables/Figures:

Most of the time table captions are not self-consistent. Please develop.

Page 17, figure 1: To help the reader about the distinction between convective and stratiform regions, please add to titles "Flight 9 – stratiform region" and "Flight 12 – convective region".

Page 19, figure 3: Where are a) and b) on the figure ? Same comment for the following figures 4-7.

Page 22, figure 8: It should be great to add on each panel the correlation and standard deviation.

---

## Referee Comment (RC2) · Anonymous Referee #2 · 22 Dec 2016

The paper "Radar Reflectivity Factors Simulations of Ice Crystal Populations from In Situ Observations for the Retrieval of Condensed Water Content in Tropical Mesoscale Convective System", by Fontaine and co-workers, presents an evaluation of a Condensed Water Content (CWC) retrieval algorithm based on W-band radar reflectivity and nearly simultaneous in-situ PSD measurements. The retrieved quantity is then compared to CWC values directly measured by another device installed on the same aircraft to get insights on the errors of the procedure.

The topic is interesting and certainly fits the scope of the Journal, discussing a measurement technique, its application and evaluation. The paper is fairly well written, even if I have the feeling that a carful language check would improve its overall quality. I suggest the publication of the manuscript, after some integration of the text, as indicated below.

In my opinion, the experimental campaign is described too loosely: no information on the scanning (or acquisition) strategy of the 94GHz radar, on the radar sampling volume, compared to the ones of other instruments. Some information on the flight path with respect to the cloud structure should also be given: it can be argued (also from Fontaine et al., 2014) that the flights where through MCS anvils, but at line 5 on page 7 "convective updrafts" are mentioned. How would impact on the method and results the presence of different ice particles, such as graupel or small hail?